# Proteomics Approach for the Discovery of Rheumatoid Arthritis Biomarkers Using Mass Spectrometry

**DOI:** 10.3390/ijms20184368

**Published:** 2019-09-05

**Authors:** Sora Mun, Jiyeong Lee, Arum Park, Hyo-Jin Kim, Yoo-Jin Lee, Hyunsong Son, Miji Shin, Mi-Kyoung Lim, Hee-Gyoo Kang

**Affiliations:** 1Department of Senior Healthcare, BK21 Plus Program, Graduate School, Eulji University, Seongnam 13135, Korea; 2Department of Biomedical Laboratory Science, College of Health Sciences, Eulji University, Seongnam 13135, Korea; 3Division of Rheumatology, Department of Medicine, Eulji University School of Medicine, Daejeon 34824, Korea

**Keywords:** rheumatoid arthritis, diagnosis, biomarker, proteomics

## Abstract

Rheumatoid arthritis is an autoimmune disease that causes serious functional loss in patients. Early and accurate diagnosis of rheumatoid arthritis may attenuate its severity. Despite a diagnosis guideline in the 2010 American College of Rheumatology (ACR)/European League Against Rheumatism (EULAR) classification criteria for rheumatoid arthritis, the practical difficulties in its diagnosis highlight the need of developing new methods for diagnosing rheumatoid arthritis. The current study aimed to identify rheumatoid arthritis diagnostic biomarkers by using a proteomics approach. Serum protein profiling was conducted using mass spectrometry, and five distinguishable biomarkers were identified therefrom. In the validation study, the five biomarkers were quantitatively verified by multiple reaction monitoring (MRM) analysis. Two proteins, namely serum amyloid A4 and vitamin D binding protein, showed high performance in distinguishing patients with rheumatoid arthritis from healthy controls. Logistic analysis was conducted to evaluate how accurately the two biomarkers distinguish patients with rheumatoid arthritis from healthy controls. The classification accuracy was 86.0% and 81.4% in patients with rheumatoid arthritis and in healthy controls, respectively. Serum amyloid A4 and vitamin D binding protein could be potential biomarkers related to the inflammatory response and joint destruction that accompany rheumatoid arthritis.

## 1. Introduction

Rheumatoid arthritis (RA) is an autoimmune disease that causes inflammation of the joints and surrounding synovial membrane [1]. Female patients are known to be vulnerable to RA, accounting for 70 to 80 percent of patients with RA. According to the World Health Organization (WHO) report, RA from musculoskeletal disorders affects approximately one percent of the world’s population and is steadily increasing at various ages [2]. Because failure of early diagnosis results in poor prognosis in patients, and the disease often persists for decades after the diagnosis, early diagnosis and appropriate drug therapy are essential for the efficient treatment of RA. In severe cases, patients are no longer able to move in daily life and eventually acquire permanent disability [3,4,5,6].

RA is diagnosed based on symptoms, blood tests, inflammatory index, and imaging tests. Patients suspected to have RA show symptoms of morning stiffness, pain, and swelling, which continue over six weeks. It is comprehensively evaluated by serological tests for rheumatoid factor (RF) and anti-citrullinated protein antibodies (ACPA) and for inflammatory markers such as CRP, an ESR test, and an imaging test [7]. However, RF-positive rate and ACPA-specificity of RA are only 60–70% and 60–75%, respectively, suggesting low diagnostic efficacy. Therefore, there has been increasing interest in identifying specific and powerful biomarkers for RA [8]; studies toward novel diagnostic biomarker discovery, using mass spectrometry, have been increasing [9,10,11,12]. Because the expression pattern of serum proteins is altered in diseased conditions, such as RA, analysis of protein expression pattern and protein function has been used for biomarker discovery. A proteomics approach using mass spectrometry can be used to determine the pattern of serum proteins and compare their absolute levels between diseased and control subjects [13]. For discovering diagnostic biomarkers, blood, urine, and body fluids such as synovial fluid may be used as the samples. Because blood is easily obtained from patients and reflects the overall status of patients, in contrast to other body fluids or urine, in our study, we obtained serum from patients with RA and tried to discover diagnostic biomarkers in informative blood samples using mass spectrometry.

Our aim was to screen patients with RA, using serum samples, for diagnostic biomarkers. In the discovery and validation set, candidate biomarkers were selected and verified. The number of patient serum samples for discovery set was 20, whereas it was 50 for the validation set (Table 1). Proteomics analysis was conducted in individual samples to reflect the characteristics of all serum samples, using a multiple reaction monitoring (MRM) method for quantitative measurement.

## 2. Results

### 2.1. Distribution of Each Group by Principal Component Analysis (PCA) and Loading Plot Analysis

Serum proteins were analyzed in healthy controls (*n* = 20) and in patients with RA (*n* = 20) by SWATH label-free quantification, and 435 proteins were identified. PCA and partial least squares-discriminant analysis (PLS-DA) were conducted to evaluate whether 435 proteins could distinguish patients with RA from healthy controls. Both PCA and PLS-DA showed 20 healthy controls and 20 patients with RA to be classified by PC1 score (Figure 1a,b). Through PC variable grouping, loading plot analysis was conducted, which showed that proteins were distributed according to the degree of expression of the identified proteins in both groups (Figure 1c). The farther the protein plot was from the center (sky blue), the better could the protein be distinguished between the two groups. Proteins with little difference in expression pattern between the two groups were concentrated in the center and are shown in dark blue (distance < 0.07) (Figure 1c).

### 2.2. Identification of Differentially Expressed Proteins (Fold-Change > 1.5, p < 0.05) and Selection of Diagnostic Biomarkers

A *t*-test analysis of the 435 identified proteins was performed using SCIEX MarkerView™ Software; 294 of them were found to be differentially expressed by more than 1.5-fold with statistical significance (*p* < 0.05). To visualize the overall alteration of protein expression pattern in groups, a heat map analysis was performed (Figure 2a). Heat map analysis showed that majority of DEPs had distinctly different expression patterns in patients with RA compared to that in healthy controls. Especially, serum protein expression patterns in patients with RA showed distinct alterations compared to that in the 10 healthy controls (HC 010–020) (Figure 2a). Volcano plot analysis showed distribution of the 294 differentially expressed proteins (DEPs) (Figure 2b). X-axis refers to log_2_ (fold-change), based on which proteins that were increased or decreased by > 1.5-fold in the X-axis were filtered. Y-axis refers to the −log_10_ (corrected *p*-value), by which proteins with *p*-value < 0.05 were filtered. Red plots on volcano satisfied both the conditions (Figure 2b). Proteins that were filtered by volcano plot analysis were analyzed to select diagnostic biomarkers for RA using individual samples in each group. For example, among the DEPs, proteins with little variation in expression patterns between samples from one group, namely between samples of healthy controls or between samples of diseased patients as well as those between samples of both groups, namely, between healthy controls and diseased patients) were selected. As per our analysis, five proteins were selected as final candidate biomarkers (Figure 2c). The selected five biomarkers had also been reported to be associated with RA, which would further help determine whether the five proteins could be potent for diagnosing RA. The five proteins were up-regulated in patients with RA.

### 2.3. Functional Analysis of Identified DEPs (Fold-Change > 1.5, p < 0.05)

Although all the 294 DEPs were not selected as candidate biomarkers, all of them are thought to affect the onset of rheumatoid arthritis by being increased or decreased in the disease environment. Otherwise, they are a result of the onset of rheumatoid arthritis. Therefore, functional analysis of the DEPs helped in understanding the pathophysiology of RA. GeneGo software was used to analyze the pathway maps, process networks, and GO processes. In a three-function analysis graph, the X-axis was -log (*p*-value). Higher the value of -log (*p*-value) in the X-axis, more statistically significant its association with DEPs was. The ranking of statistically significant functions has been presented on the Y-axis. Pathway maps analysis showed blood coagulation, lectin-induced complement pathway (immune response), classical complement pathway (immune response), alternative complement pathway (immune response), and lipoprotein metabolism to be associated with DEPs. Among them, blood coagulation had a strong and distinct relationship with proteins that increased or decreased in RA (Figure 3a). In addition, process networks analysis showed blood coagulation, inflammation-related complement system, inflammation-related kallikrein-kinin system, proteolysis-related ECM remodeling, and cell adhesion-related platelet-endothelium-leucocyte interactions to be related to DEPs (Figure 3b). Finally, GO processes showed the processing of antigen and presentation of exogenous peptide antigen via MHC class I, TAP-independent regulation of immune response, defense response, innate immune response, and type I interferon signaling pathway to be related to DEPs (Figure 3c).

### 2.4. Selection of Target Peptide for MRM Analysis

MRM analysis for biomarker candidates, including complement C3, kallistatin, vitamin D binding protein, serum amyloid A4 protein, and angiotensinogen was conducted. Based on the SWATH-quantitative analysis of peptides, corresponding to the final biomarker proteins in discovery set, target peptides for MRM analysis were selected: (1) ISLPESLK (complement C3); (2) LGFTDLFS (kallistatin); (3) ALQDQLVLVAAK (vitamin D binding protein); (4) THLPEVFLSK (serum amyloid A4 protein); and (5) ALQDQLVLVAAK (angiotensinogen). Information about the transition of selected peptides was obtained using Skyline software (Table 2). The highly sensitive peptide in SWATH acquisition was selected first. The final selected peptide satisfied the following conditions: (1) peptide had no mis-cleavage site; (2) unmodified peptide; (3) not including M in the sequence; (4) having 7–15 peptide sequence; and (5) having low FDR (usually 0).

### 2.5. MRM Analysis and Multivariate Analysis of the Multi-Marker Panel Using Individual Serum Samples

The five biomarkers for RA were confirmed by MRM analysis using samples from healthy controls and patients with RA. The five proteins were analyzed with 5 μL of each patient sample using UPLC-MRM/MS. For MRM validation, one peptide and three Q3 ions were selected per target protein. Among the three Q3 ions, the peak with the highest sensitivity was used for quantification. MRM analysis showed that the five proteins distinguished between healthy controls and patients with RA with high sensitivity and specificity. Whereas complement C3 and kallistatin were down-regulated in RA, vitamin D-binding protein, serum amyloid A4 protein, and angiotensinogen were up-regulated (data not shown). Among the five biomarker proteins, serum amyloid A4 protein and vitamin D-binding protein had AUC > 0.8. AUC of serum amyloid A4 protein and vitamin D-binding protein were 0.8307 and 0.8502, respectively (Figure 4). To evaluate the diagnostic efficacy of the two proteins, logistic analysis was conducted. Classification accuracy was 86.0% and 81.4% in patients with RA and in healthy controls, respectively (Figure 5).

## 3. Discussion

To identify serum biomarkers in patients with RA, serum protein profiling was conducted individually with 43 serum samples from the normal group and 50 from the patient group with RA. Analysis of serum proteins revealed five proteins in the discovery set that could be considered as potential candidate biomarkers; two of the five candidate biomarkers, namely serum amyloid A4 and vitamin D-binding protein, had higher efficiency than the AUC value of 0.8. Although analysis of the pooled samples requires less time and had lower cost for the sample preparatory step and MS analysis, it provides insufficient information for concrete and specific conclusions. Because the amount of serum protein varies across individuals, the protein expression pattern in a certain patient might represent the overall state of all persons in the group. In addition, individually analyzed samples can be further analyzed based on the corresponding clinical data. Therefore, in our study, serum samples were analyzed individually, based on which we could select and verify the two potential biomarkers, namely serum amyloid A4 and vitamin D-binding protein.

Functional analysis revealed blood coagulation as the most significant parameter in the pathway map as well as process network function analysis. Of the proteins involved in the blood coagulation pathways, 21 were found to be enhanced in the patient group compared to the control group. All 21 proteins levels increased by more than 1.8-fold, and the increase was statistically significant (*p*-value less than 0.05). The results therefore demonstrated the activation of blood coagulation in patients with RA. Likewise, in a previous study, coagulation had been reported to be abnormally activated by the altered expression of coagulation-related factors in patients with RA and aggravated RA [14,15,16,17]. For example, the overexpression of fibrinogen in serum, increase in platelets, and activity of plasmin had been reported in relation to RA. In this study, functional analysis showed that serum proteins from patients with RA were maximally associated with blood coagulation. In both pathway maps and process networks function analyses, the complement system was confirmed to be activated in patients with RA compared to that in the control group. The complement reaction is activated by three main pathways, mediated by differentially expressed serum proteins and membrane-related proteins [18,19,20]. The three main pathways include the lectin-induced complement pathway, classical complement pathway, and alternative complement pathway. In this study, we confirmed that the DEPs identified in patients with RA were involved in all the three major pathways. Previous studies had also confirmed the complement pathway to be activated in patients with RA. Besides, as the consumption of complement protein increased, its level was found to decrease in the synovial fluid [18].

VDBP is known as an actin-scavenging protein. When tissues are damaged in RA, cells with increased permeability secrete F-actin, which corresponds to polymerized actin [21,22,23]. As a result, blood vessels are blocked, leading to pathological conditions such as microthrombosis. When the tissue is damaged, vitamin D-binding protein, circulating in the blood, gets transferred to the damaged tissue, from the blood vessel, and combines with F-actin to allow the removal of actin [23]. Briefly, plasma VDBP and F-actin complexes are formed, subsequently changing the polymer F-actin into monomer. Thereafter, F-actin is combined with GC-globulin and removed from the liver [24]. Thus, when destruction of synovial membrane and cartilage tissue is processed in RA, VDBP is suggested to move to the synovial membrane and synovial fluid and remove actin [25]. Thus, our results implied that increased VDBP migration to the tissue may result in loss of protein from the blood. However, in joints, increased migration of VDBP to tissues suggests overexpression of the protein according to relevant needs in the certain tissue. Vitamin D2 and D3 in our body are made by sunlight or food uptake. Both vitamin D2 and D3 circulate in the blood and get transferred to the liver by VDBP. They are converted into 25-hydroxy vitamin D2 before moving to the kidney, after which they are converted into 1,25-dihydroxy vitamin D3 and mediate immune cells to act as anti-inflammatory agent in the bone tissue, intestines, and immune cells [26,27]. When RA is active, vitamin D may be assumed to be carried by VDBP to various tissues to suppress the inflammatory response. Vitamin D-binding protein may also be suggested to act as a defense mechanism against in vivo changes caused by the progression of RA, which is consistent with the role of vitamin D-binding proteins as actin-scavenging proteins.

SAA4 is an acute-phase protein, known to increase rapidly during inflammatory reactions. In RA, SAA4 is excessively secreted in association with the action of cytokine phosphorus, which is secreted by the immune cells. Owing to this, SAA4 is widely known as an acute reactant. SAA4 is as sensitive as CRP for the diagnosis of RA. Previous studies conducted in our laboratory had consistently reported SAA4 to be a potential candidate biomarker of RA, which is being verified by increasing number of validation samples. In our previous study, SAA4 had been proven to be potent not only for diagnosing RA, but also for monitoring the disease activity of RA. Therefore, if the candidate biomarkers are verified using large samples, they will be able to play an important and powerful role as a diagnostic as well as a monitoring marker for RA.

Proteomics approach has been used to discover biomarkers in previous studies, wherein usually a single marker was used to represent a particular disease [11,28]; however, in this study, we present a multi-marker panel to increase the effectiveness of diagnosis of RA. As opposed to our study, in previous studies experiments were conducted by pooling patient samples at the discovery stage. Recently, scientists have tried to discover biomarkers through individual samples; individual sample analysis is not only expensive, but it creates difficulty in discovering biomarkers. More than a few dozen sample groups were used for analysis with less than 20 people in each group [11,28,29]. In this study, the patients, *n* = 20 in the discovery set and *n* = 50 in the validation set, were analyzed individually to enable the selection of a candidate marker. Individual analysis of a large number of samples is more useful in confirming the expression value of the proteins that are aggregated in the same group and proteins that are distinct among different groups. For discovery of markers for diagnosis of RA, a group of patients with other autoimmune diseases or non-rheumatic arthritis diseases as a control group along with a healthy control is essential [25]; however, we did not analyze patients with other autoimmune diseases or non-rheumatic arthritis, such as osteoarthritis, psoriatic arthritis as a control group, as the major purpose of this study was to differentiate healthy individuals from patients with rheumatoid arthritis. Therefore, for a clear diagnosis, the study was conducted by comparing healthy individuals and patients with rheumatoid arthritis. Our results should be further interpreted by analyzing patients with other autoimmune diseases or non-rheumatoid arthritis for practical application of these biomarkers in the future.

This study was conducted on individuals aged between 50 and 70 years. Although RA incidence is high in the older population, RA also occurs in young individuals. Therefore, future studies should focus on both older as well as younger population. In order to verify the target biomarker proteins, one of the most sensitive peptide was selected and its absolute quantification was conducted by MRM analysis. Further verification with one or more peptides of the target proteins needs to be conducted for the use of the identified biomarkers in the clinical field. The range of expression levels of SAA4 and VDBP was wide in rheumatoid arthritis patients, although the expression of these biomarkers was higher in the rheumatoid arthritis patients than in the control group (Figure 4). We observed that the biomarkers discovered in this study have higher accuracy than RF, which is an RA diagnostic marker currently used in the clinical field (data not shown), but the accuracy of these markers (81.4%) still needs to be improved for use as clinical markers in the future. In our previous study, we observed that the AUC value for RF was only 0.6477 (the AUC value of the multi-biomarker set presented in the current study was 0.9) [12]. Thus, markers more efficient than the conventional markers (RF) are needed for efficient diagnosis of RA. In future studies, it is necessary to evaluate the efficiency and usefulness of the markers that are presented in this study through detailed classification and analyses of confounding factors.

When RA occurs, VDBP acts readily to maintain homeostasis of joint tissues; SAA4 also immediately responds to the inflammatory reactions by inducing production of cytokines. Therefore, VDBP and SAA4 proteins are suggested as the potential diagnostic markers that represent inflammation and joint destruction caused by the inflammatory reactions in RA.

## 4. Materials and Methods

### 4.1. Healthy controls and Patients

Serum samples for biomarker discovery were obtained from the Eulji University Hospital Institutional Review Board (EMC 2016-03-019, 31th March 2016). The number of serum samples taken from patients with RA and from healthy controls was 20 each in the discovery set and 50 and 43, respectively, in the validation set. The clinical information of the subjects is presented in Table 1. All the patients were treated with methotrexate, hydroxychloroquine, salazopyrine, leflunomide, cyclosporine and misoribine. Blood from both, patients with RA and healthy controls, was collected in vacutainers without anticoagulant. Serum was separated from blood at 4000× *g* for 5 min and stored at −80 °C until MS analysis.

### 4.2. Depletion of Highly Abundant Serum Proteins

To eliminate albumin, IgG, antitrypsin, IgA, transferrin, and haptoglobin that were expressed the most in human serum, a multiple-affinity removal system liquid chromatography (LC) column (human 6-HC, 4.6 × 50 mm; Agilent Technologies, Santa Clara, CA, USA) was used. Briefly, serum samples loaded onto a multiple-affinity removal system LC column were eluted into fractions containing low-abundance proteins while removing the highly abundant ones. The eluted product was used for MS analysis.

### 4.3. Information-Dependent Acquisition (IDA) by SCIEX TripleTOF 5600

Concentration of serum proteins was measured and 100 μg protein samples were prepared for mass analysis. Proteins were reduced by treatment with 5 mM Tris (2-carboxyethyl) phosphine (Pierce, Rockford, IL, USA) at 37 °C, 300 rpm, for 30 min. For alkylation, 15 mM Iodoacetamide (Sigma-Aldrich, St. Louis, MO, USA) was added to the samples at 25 °C, with agitation at 300 rpm for 1 h in the dark. Proteins were cleaved into peptides, using trypsin, overnight at 37 °C. Mass spectrometry-grade trypsin gold (Promega, Madison, WI, USA) was used for the purpose. After a sample containing chemical reagents was cleaned by a C18 cartridge (Waters, Milford, MA, USA), it was separated based on isoelectric point using the OFFGEL Fractionator with a 12-well setup (3100 OFFGEL Low Res Kit, pH 3–10; Agilent Technologies, Santa Clara, CA, USA). To identify the proteins, the peptide fraction that was separated into 12 samples was analyzed using a TripleTOF 5600 mass spectrometer (AB SCIEX, Concord, ON, Canada) combined with an Eksigent nanoLC 400 system and cHiPLC^®^ (AB SCIEX, Concord, ON, Canada).

### 4.4. Relative Quantification and Data Processing by SCIEX TripleTOF 5600

For relative quantification of each sample, a 2 µL sample containing 100 µg/mL was injected onto an Eksigent ChromXP nanoLC trap column (350 µm i.d. × 0.5 mm, ChromXP C18 3 µm) at a flow rate of 5000 nL/min for 5 min and was eluted from the Eksigent ChromXP nanoLC column (75 µm i.d. × 15 cm). Flow rate was 300 nL/min and the gradient of mobile phase B was 5–90%. Total run time was 120 min. The following gradient method was used: (time/% B) 0 min/5% mobile phase B, 10.5 min/40%, 105.5 min/90%, 111.5 min/90%, 112 min/5%, and 120 min/5%. Mobile phase B was composed of 100% ACN/0.1% formic acid in HPLC-grade water, and mobile phase A was composed of 0.1% formic acid in HPLC-grade water. The mass-to-charge ratio (*m*/*z*) in MS scan range and MS/MS scan ranges were *m*/*z* 250-2500 and *m*/*z* 100-2500, respectively.

### 4.5. Absolute Quantification and Data Processing by SCIEX QTAP 5500

For the determination of MRM Q1/Q3 ion pairs, Skyline was used (http://proteome.gs.washington.edu/software/skyline). The peptide sequence was imported into Skyline, which then selected a precursor (Q1) that was a double-charged peptide with three fragment ions per precursor. Collision energy was determined in a direct infusion experiment using a mixture of candidate peptides. Optimization of MRM analysis was performed with both Q1 and Q3 sets using MRM scanning. CE, DP, CE, and CXP, for each transition, were determined by compound optimization (Table 1). A SCIEX Exion LC and QTRAP 5500 were used to analyze the serum samples from healthy controls and from patients with RA. Five-microliter samples were loaded onto an ACQUITY UPLC BEH C18 column (130 Å, 1.7 µm, 2.1 mm × 150 mm), followed by an ACQUITY UPLC BEH C18 VanGuard pre-column (130 Å, 1.7 µm, 2.1 mm × 5 mm), using a 250 μL/min flow rate and a gradient from 5-90% mobile phase B over a 30-min total run time. The following gradient method was used: (time/% B) 1 min/5% mobile phase B, 50 min/40%, 21–25 min/90%, and 25.5–30 min/5%. Mobile phase B was composed of 0.1% formic acid in HPLC-grade ACN, whereas mobile phase A was composed of 0.1% formic acid in HPLC-grade water. Source parameters for the acquisition method were as follows: curtain gas 30 psi, low collision gas, ion spray voltage 5500 V, temperature 400 °C, ion source gas 1 (GS1) 40 psi, ion source gas 2 (GS2) 60 psi. Three ion pairs per peptide were used for the quantification of target peptides. The final quantification was conducted using a quantifier ion pair per peptide. For standard curve, tryptic peptides were synthesized with >70% purity. Two-fold serial dilutions from 1 mM standard stock were performed in 0.1% formic acid or DMSO, following the manufacturer’s instructions.

### 4.6. Statistical Analysis

We conducted independent *t*-tests for relative and absolute quantification between healthy controls and patients with RA; a *p-*value less than 0.05 was defined as statistically significant.

## Figures and Tables

**Figure 1 ijms-20-04368-f001:**
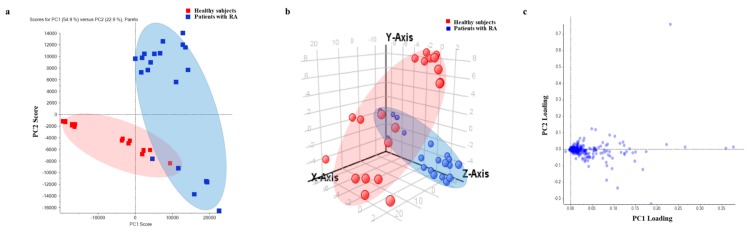
Protein quantification by SWATH acquisition and PCA for group clustering. (**a**) PCA showed 54.9% of the proteins (PC1) to be divided between healthy controls and patients with RA (vertical line). The plot represents the individual samples. Red and blue dots represent healthy controls and patients with RA, respectively. (**b**) Partial least squares-discriminant analysis (PLS-DA) showed the patient group with RA to be separated from healthy controls. (**c**) PC variable grouping based on expression pattern in healthy controls and patients with RA.

**Figure 2 ijms-20-04368-f002:**
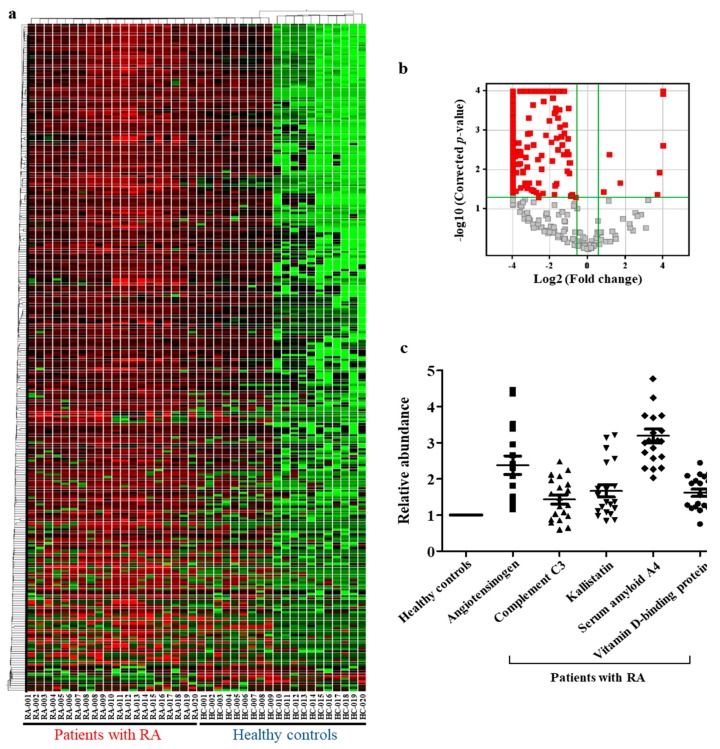
Visualization of differentially expressed proteins (DEPs, by more than 1.5-fold) and selected biomarker candidates by SWATH acquisition. (**a**) Cluster analysis of DEPs (more than 1.5-fold with statistical significance). (**b**) Volcano plot analysis of DEPs (more than 1.5-fold with statistical significance). (**c**) Relative expression of selected biomarker candidates in patients with RA compared to that in healthy controls. Abundance of the five proteins in patients with RA was normalized to that in healthy controls.

**Figure 3 ijms-20-04368-f003:**
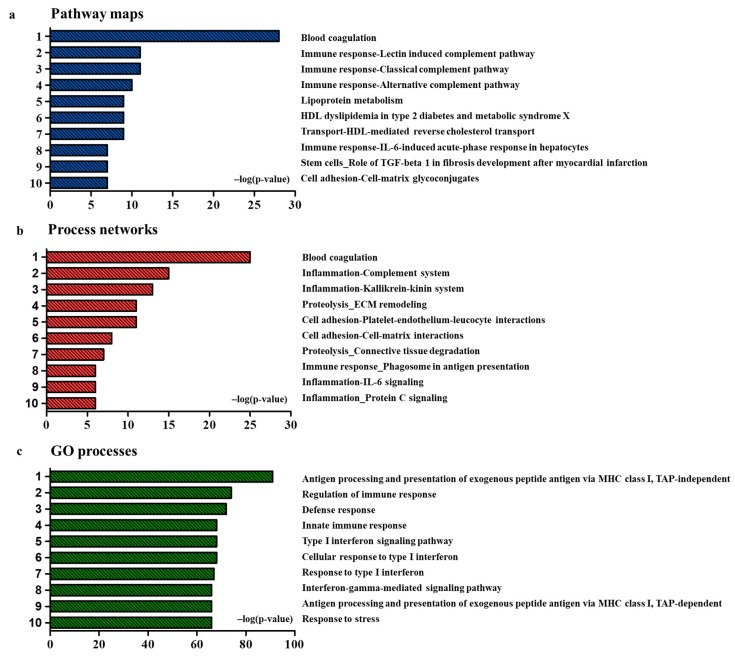
Pathway maps, process networks, and GO processes associated with proteins differentially expressed between healthy controls and patients with RA. (**a**) Pathway maps significantly associated with proteins differentially expressed between healthy controls and patients with RA. The pathway map with the lowest *p*-value was of blood coagulation. (**b**) Process networks significantly associated with proteins differentially expressed between healthy controls and patients with RA. Process network with the lowest *p*-value was of blood coagulation. (**c**) GO processes significantly associated with proteins differentially expressed between healthy controls and patients with RA. GO process with the lowest *p*-value was of antigen processing and presentation of exogenous peptide antigen via MHC class I, TAP-independent.

**Figure 4 ijms-20-04368-f004:**
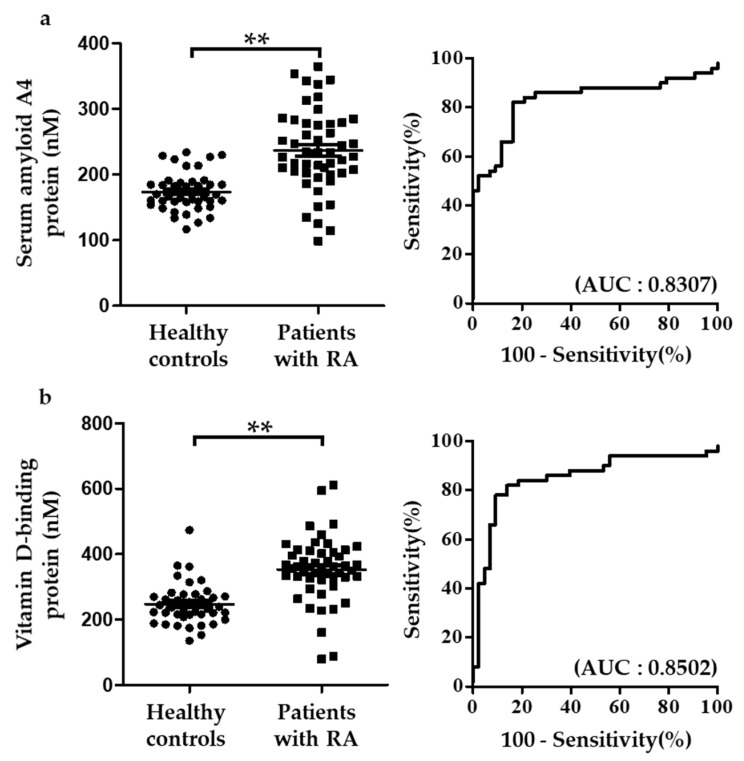
Dot plots and ROC curve of selected biomarker candidates in healthy controls and patients with RA. Proteins, significantly altered in patients with RA than in healthy controls, were selected. (**a**,**b**) Serum amyloid A4 protein and vitamin D-binding protein were compared between healthy controls and patients with RA. The number of healthy controls and patients with RA was 43 and 50, respectively. Plots indicate individual protein abundance of each group. Data are presented as mean ± SEM. Independent *t*-tests were used to determine statistical significance. ** *p* < 0.001.

**Figure 5 ijms-20-04368-f005:**
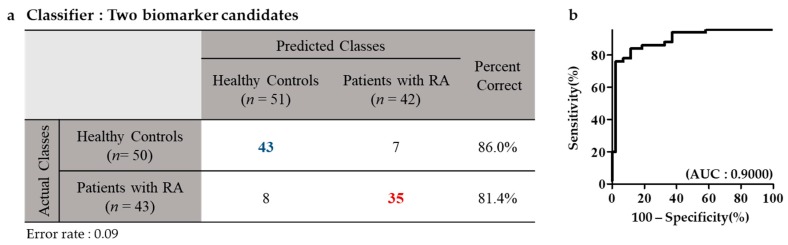
Logistic analysis of selected biomarker candidates in healthy controls and patients with RA. (**a**,**b**) The number of healthy controls and patients with RA for logistic analysis was 43 and 50, respectively. Classification accuracy was 86.0% and 81.4% in healthy controls and in patients with RA, respectively.

**Table 1 ijms-20-04368-t001:** Demographics of the healthy controls and the patients with rheumatoid arthritis (RA).

Variables	Discovery Set (LC-MS/MS)		Validation Set (MRM)	
	Healthy Controls (*n* = 20)	RA Patients (*n* = 20)	Healthy Controls (*n* = 43)	RA Patients (*n* = 50)
Sex (Female/Male)	14/6	14/6	25/18	39/11
Age (Years)	55.3 ± 3.9	59.2 ± 5.8	56.9 ± 4.7	59.9 ± 6.7
RF (IU/mL)	-	97.95 ± 81.1	-	79.5 ± 67.4
RF-Positive, *n*	-	16	-	40
RF-Negative, *n*	-	4	-	10
ACPA (U/mL)	-	161.2 ± 120.5	-	124.8 ± 112.6
ACPA-Positive, *n*	-	15	-	35
ACPA-Negative, *n*	-	5	-	15
DAS28	-	3.3 ± 1.2	-	2.7 ± 1.2
Low activity, *n* (DAS28 < 3.2)	-	12	-	37
Moderate activity, *n* (3.2 ≤ DAS28 > 5.1)	-	6	-	9
High activity, *n* (DAS28 > 5.1)	-	2	-	3

LC-MS/MS, Liquid chromatography–tandem mass spectrometry; MRM, multiple reaction monitoring; RF, rheumatoid factor; ACPA, anti-citrullinated protein antibodies; DAS28, Disease activity score in 28 joints.

**Table 2 ijms-20-04368-t002:** List of the 13 target peptides and their parameters for multiple reaction monitoring (MRM).

Compound Name	Gene Name	Peptide Sequence	Q1 (*m*/*z*)	Q3 (*m*/*z*)	Q3 Ion Type	Q3 Ion Charge	DP (volts)	CE (volts)	CXP (volts)
Angiotensinogen	*ANGT*	ALQDQLVLVAAK	634.882	956.578	Y9	2	77.4	31.7	11
				600.408	y6	2	77.4	31.7	11
				289.187	Y3	2	77.4	31.7	11
Complement C3	*CO3*	ISLPESLK	443.776	573.3	y5	2	61	19	28
				686.3	y6	2	61	17	30
				773.3	y7	2	61	19	40
Kallistatin	*KAIN*	LGFTDLFSK	514.3	609.3	y5	2	66	23	42
				710.3	y6	2	66	23	36
				857.4	y7	2	66	21	42
Serum amyloid A4 protein	*SAA4*	FRPDGLPK	465.3	516.2	b4	2	65	25	32
				573.2	b5	2	65	27	14
				244.1	y2	2	65	27	14
Vitamin D-binding protein	*VTDB*	THLPEVFLSK	585.83	819.461	y7	2	73.8	29.9	11
				239.114	b2	2	73.8	29.9	11
				352.198	b3	2	73.8	29.9	11

DP: Declustering potential; CE: Collision energy; CXP: Collision exit potential.

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
