# Peer review of "Proteomics Approach for the Discovery of Rheumatoid Arthritis Biomarkers Using Mass Spectrometry"

_ijms, 2019, doi:10.3390/ijms20184368_

Round 1

Reviewer 1 Report

Presented manuscript is an extensive study dealing with an interesting and important topic. Overall, the article is well written, the authors presented many results. However, I have some comments to the authors:

- the informations regarding the subjects used in the study are not complete, the data on age, gender, etc. are missing,

- similarly, the authors did not provide information regarding the stage of the disease in which the patients were by the sample collection, this is very important and should be taken into consideration by the searching for novel biomarkers,

- although the authors stated in the setting of the goals of the study that 20 serum samples were used for the discovery study, and they used 50 samples for the validation set, however, this is not clearly described in the methods,

- from the figure 4 in the form of dot plots is evident that although the means of SAA4 and VDBP are higher in patients with rheumtoid arthritis, the range of values is wide, but this is not mentioned in the results,

- the authors correctly stated that further studies would be helpful for the evaluation of the usefulness of these biomarkers for early diagnosis of rheumatoid arthritis in the clinical field.

Reviewer 2 Report

The paper is of interest and the results are of interest.

Major points

The authors must describe better the patients' characteristics used for their analysis. Did they select untreated, newly diagnosed patients?

What was their autoantibody status, radiological, biochemical and clinical profile?

Did they select early, very-early or well-established patients with RA?

it is very important this, as patients under treatment or of distinct radiological/clinical profile may have different levels of measured proteins.

The most important characteristic of biomarkers is the ability to stratify patients with rheumatoid arthritis, from patients with other autoimmune rheumatic or non diseases (ostearthritis, psoriatic arthritis, spondylarthropathies etc). The authors must discuss why they did not select pthological cohorts in their analysis and just picked up healthy controls.

The authors must discuss other studies using proteomic analysis using similar approaches and discuss in detail (differences, similarities) amongst these studies and their study.
